# End-of-life care for people with severe mental illness: mixed methods systematic review and thematic synthesis of published case studies (the MENLOC study)

Michael Coffey [1], Deborah Edwards [2], Sally Anstey [2], Paul Gill [2], Mala Mann [3], Alan Meudell [4], Ben Hannigan [2]

¹School of Health and Social Care, Swansea University, Swansea, UK
²School of Healthcare Sciences, Cardiff University, Cardiff, UK
³University Library Services, Cardiff University, Cardiff, UK
⁴Service User Consultant, Cardiff, UK

**Correspondence to**
Professor Michael Coffey;
m.j.coffey@swansea.ac.uk

## ABSTRACT

**Objectives** People with severe mental illness (SMI) have significant comorbidities and reduced life expectancy. The objective of the review reported in this paper was to synthesise material from case studies relating to the organisation, provision and receipt of care for people with SMI who have an end-of-life (EoL) diagnosis.

**Design** Systematic review and thematic synthesis.

**Data sources** MEDLINE, PsycINFO, EMBASE, HMIC, AMED, CINAHL, CENTRAL, ASSIA, DARE and Web of Science from inception to December 2019. Supplementary searching for additional material including grey literature along with 62 organisational websites.

**Results** Of the 11 904 citations retrieved, 42 papers reporting 51 case studies were identified and are reported here. Twenty-five of the forty-two case study papers met seven, or more quality criteria, with eight meeting half or less. Attributes of case study subjects included that just over half were men, had a mean age of 55 years, psychotic illnesses dominated and the EoL condition was in most cases a cancer. Analysis generated themes as follows diagnostic delay and overshadowing, decision capacity and dilemmas, medical futility, individuals and their networks, care provision.

**Conclusions** In the absence of high-quality intervention studies, this evidence synthesis indicates that cross disciplinary care is supported within the context of established therapeutic relationships. Attention to potential delay and diagnostic overshadowing is required in care provision. The values and preferences of individuals with severe mental illness experiencing an end-of-life condition should be recognised.

**PROSPERO registration number** CRD42018108988.

## INTRODUCTION

Mental ill health is the leading cause of years lived with disability in 56 countries and the second leading cause in a further 56.[1] The wider economic costs of mental illness in England have been estimated at £105.2 billion each year[2] and £7.2 billion in Wales.[3]

### Strengths and limitations of this study

► This systematic review represents, to our knowledge, the first synthesis of case studies in end-of-life care for people with prior severe mental illness.

► Case studies have a long history in medicine, provide useful insights and fill an important gap in the absence of intervention research.

► We identified 51 case studies across 42 articles with 25 meeting 7 or more of 8 quality criteria.

► Most papers were from high-income countries, and thus limiting transferability.

► Future research should focus on generating high-quality intervention studies in this field.

The term 'severe' mental illness (SMI) has longstanding currency within the fields of mental health policy, services and practice.[4] It continues to be used in research,[5] and by the National Institute for Health Research (NIHR) Dissemination Centre.[6] People with SMI have high comorbidities,[7] and higher mortality rates and reduced life expectancy compared with the general population[8] across all age groups,[9] with a 10–20-year reduction in life expectancy.[10] Inequities, not limited to care at the end of life specifically, can be explained with reference to individual and system-level factors. People with SMI are less likely to attend health screenings and may respond to symptoms differently.[11] They may delay or avoid help, and are more likely to exhibit disruptive behaviours or miss contacts with health professionals[12 13] putting them at risk of delayed disease detection.[7] Inadequate support systems are also common among those with SMI, affecting their ability to access appropriate clinical care and navigate complex health systems.[14] Other factors

influencing variations in mortality and morbidity for people living with SMI include poor previous experiences of seeking help from healthcare professionals, incorrect attribution of physical symptoms to psychiatric disorder by care staff and lack of experience by mental health professionals in determining how and when to refer onwards to other appropriate services.[12 15]

End-of-life (EoL) care refers to the care of people with diagnoses of advanced, incurable, cancer and/or end-stage lung, heart, renal or liver failure and who are likely to die within the next 12 months.[16] It includes care provided in hospitals, hospices and other institutional settings (such as care homes, prisons, and hostels) and care provided at home and via outreach to people who may also be homeless.

Although cancer incidence among people living with SMI is similar to that of the general population, mortality rates are double.[17 18] This disparity may be related in part to late presentation and reduced use of interventions such as surgery, systemic anticancer therapies or radiotherapy.[19 20] The experience of SMI can delay access, detection and treatment of life-threatening physical disorders and their symptoms, specifically pain.[21] Consequently, this patient cohort is more likely to present advanced cancers that are more complex and costly to treat, being less likely to undergo invasive treatments and more likely to die.[22]

Research at the interface of physical and mental healthcare is recognised as a priority.[23] Policies focus on improving EoL care, where diagnosis is immaterial.[24–32] These policies require the introduction of palliative and supportive care earlier in the illness trajectory. In national policy, the needs of people with SMI who develop advanced incurable cancer and/or end-stage lung, heart, renal or liver failure are acknowledged poorly, or not at all. This group faces the prospect of 'disadvantaged dying',[33 34] at a time when quality of care in the last months of life should be uniformly high for all groups.

Our work is distinguished from other work in this field as we adopt an inclusive approach to consider policy papers, primary research, and in this paper we report the inclusion of published case studies as a further part of our NIHR-funded systematic review focused on published evidence for EoL care for people with SMI.[35] For the purposes of this review, all descriptions of the care and experiences of individuals with pre-existing SMI and EoL diagnosis identified via our search strategy were included. Case studies were excluded if no SMI condition was identified, if no EoL condition was identified or if the mental illness was not pre-existing.

Case studies have a long tradition in healthcare[36] aiming to provide insights from the specific, to illustrate broader lessons.[37] Given the limited research into EoL care for people experiencing SMI, a synthesis of case studies can contribute new understandings and direct us towards patterns of care that might otherwise remain hidden.

## METHODS

The protocol is registered in PROSPERO (see online supplemental file 1) and followed the Preferred Reporting Items for Systematic Review and Meta-Analysis (PRISMA)[38] (see online supplemental file 2). The methods are reported in detail in the full project report[35] and summarised here.

### Inclusion criteria
#### Population

Adult participants (>18 years of age) with SMI who have an additional diagnosis of advanced, incurable, cancer and/or end-stage lung, heart, renal or liver failure and who are likely to die within the next 12 months were considered. SMI was defined as including those with, but not limited to, schizophrenia, schizophrenia spectrum and other psychotic disorders, schizotypal and delusional disorders, bipolar affective disorder, bipolar and related disorders, major depressive disorder and disorders of adult personality and behaviour. SMI is an imprecise term, and definitions also include duration and disability as criteria.[4 6] Therefore, we additionally included studies of enduring conditions such as post-traumatic stress disorder (PTSD) and anorexia nervosa where our searches located them.[35]

#### Types of intervention and phenomena of interest
EoL care.

### Context

EoL care provided in hospitals, hospices and other institutional settings (such as care homes, prisons and hostels), and care provided in the home and via outreach to people who may also be homeless.

#### Types of evidence
Case studies.

### Exclusion criteria
Evidence relating to:
► Mental health problems (eg, depression) as a consequence of terminal illness (eg, cancer or chronic organ failure).
► EoL care for people with mental and behavioural disorders due to psychoactive substance use, except where these coexisted with SMI as specified above.
► EoL care for people with dementia or other neurodegenerative diseases, except where these coexisted with SMI as specified above.

In this review, we specifically focused on gathering evidence of EoL care for people with pre-existing SMI who would, broadly put, have used secondary mental health services. We acknowledge the importance of EoL care for people with neurodegenerative disorders, frailty and other conditions but recognise these as a separate population, and in the interests of achieving a more focused review we excluded material in these areas.

The search strategy was developed for Ovid MEDLINE and adapted for the other databases (see online supplemental file 3). Searches were run on the following

databases from inception for studies published in the English language.

- ► MEDLINE ALL; EMBASE; HMIC, PsycINFO; AMED;
- ► CINAHL; CENTRAL;
- ► ASSIA;
- ► DARE; Web of Science (WoS).

The keywords that were used to inform these searches included the following:

Palliative care OR Hospice care OR Terminal care OR Terminally ill OR End of life care OR Last year of life

AND

Neoplasms OR Cancer OR heart failure, lung failure, liver failure or renal failure

AND

Mental health OR Depression OR Mental disorders OR Depressive disorder OR Personality disorders OR Bipolar disorder OR Schizophrenia OR Mental illness.

Supplementary searches were conducted for additional papers, information on studies in progress, unpublished research, research reported in the grey literature and personal blogs (see online supplemental file 4).

Searches were also conducted using Google as described by Mahood *et al*.[39 40] The first 10 pages of each Google output were screened using the terms:

- ► 'Palliative care' and 'mental illness'.
- ► 'End of life' and 'mental illness'.
- ► "End of life" and schizophrenia (searching first 5 pages of output).
- ► 'End of life' and bipolar (searching first five pages of output).

The contents of the last 2 years of the *Journal of Pain and Symptom Management, Cancer, Psycho-Oncology* and *BMJ Supportive & Palliative Care* were hand searched. These journals were selected due to the large number of outputs identified in database searches being published within them. Reference lists of included studies were scanned, and forward citation tracking performed using WoS.

## Screening

References were collected and deduplicated using Covidence systematic review software (Veritas Health Innovation, Melbourne, Australia).

## Data extraction

Demographic data were extracted into tables and followed the format recommended by the Centre for Reviews and Dissemination (CRD).[40] These data were checked by a second reviewer independently for accuracy and completeness. Where multiple publications from the same study were identified, data were extracted and reported as a single study.

## Data synthesis

All case studies were available in full-text form, were read and re-read, uploaded to NVivo, inductively coded by one reviewer and checked by a second, and then synthesised into five themes.[41]

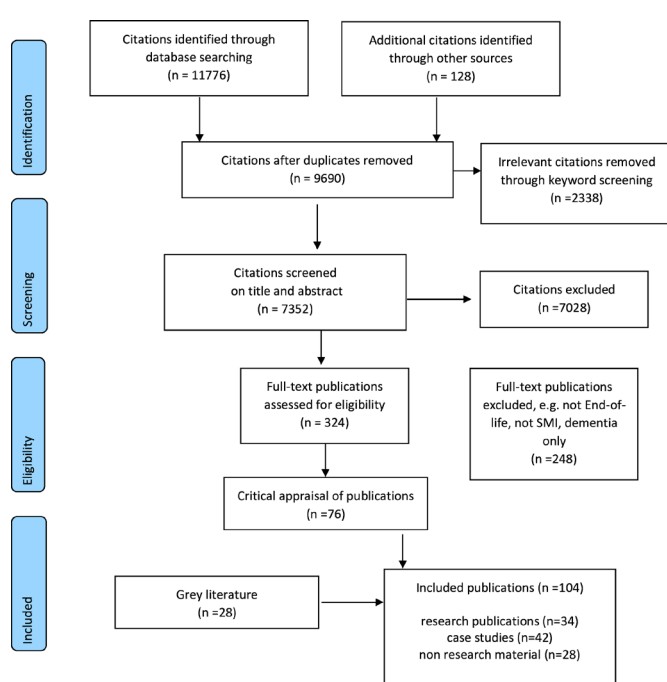

**Figure 1** Preferred Reporting Items for Systematic Review and Meta-Analysis flow chart. SMI, severe mental illness.

## Assessment of methodological quality

The Joanna Briggs Institute (JBI) critical appraisal checklist for case reports was used.[42] This is an eight-item checklist ('yes', 'no', 'unclear', 'not applicable') from which an overall score is generated reflecting the number of items answered 'yes' (see online supplemental file 5). Items related to risk of bias, adequate reporting and statistical analysis.

## Patient and public involvement

This study included public and patient involvement perspectives commencing with original concept, the definition of search terms and parameters, study steering group, and onwards to impact and dissemination (eg, coauthor on report and papers).

## RESULTS
### Search results
Of the 11904 citations retrieved, 42 case study papers were identified and are reported in this current paper. The PRISMA flow diagram is shown in figure 1.

### Description of case studies
There were 42 publications containing 51 case studies of individuals with an existing mental illness diagnosis who went on to develop an EoL condition (see online supplemental table 1). In four case studies, the purpose of the paper was to show the application of a particular model of care for example, dynamic system analysis (DSA),[43] or stepwise psychosocial palliative care (SPPC).[44 45] Case studies were mostly published in peer-reviewed research journals (n=38) with two conference abstracts,[46 47] one appearing in a report[48] and one a first-person account

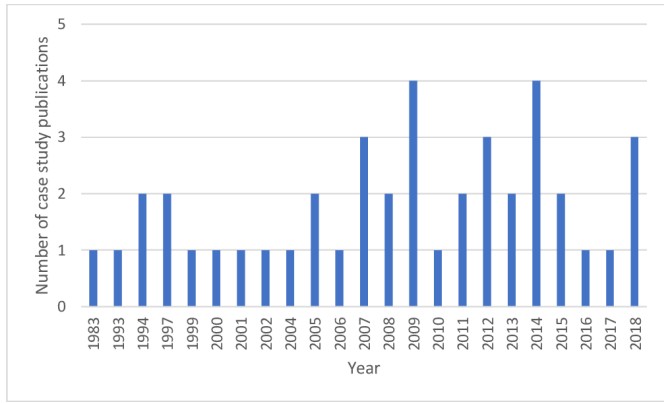

**Figure 2** Histogram of year of publication of case studies.

blog.[49] The case studies ranged in depth from discursive papers with little direct detail about the individuals involved, to those who focused mainly on the patients' physical illness. Case studies reported on the health and/or social care provided to individuals, and as such provided insights into everyday practice and its associated opportunities, challenges and dilemmas.

### Country of research
Papers were published in the USA (n=28)[20 44 45 49–72]; the UK (n=4)[48 73–75]; Canada (n=3)[76–78]; Australia (n=1)[9]; France (n=1)[79]; Israel (n=1)[80]; Mexico (n=1)[47]; Netherlands (n=1)[43]; Singapore (n=1)[81]; with one conference abstract not stating a country.[46]

### Year of publication
The earliest case study was published in 1983[60] and outputs appeared regularly up until the end of the search period in 2019. Figure 2 displays the year of publication of the 42 case study articles.

### Attributes of service users
The age range across the case studies was 20–91 years (mean 55 years) and was provided for all except two.[49 70] Women were the focus of 24 case studies. Diagnosis was reported for all case study individuals and included psychotic-type diagnoses (schizophrenia, psychosis, schizoaffective and bipolar conditions) (31 publications reporting 35 case studies),[9 20 43 46–49 52 55–57 59 60 62–74 76–82] personality disorder (5 publications reporting 5 case studies),[50 51 53 58 59] PTSD (3 publications reporting 3 case studies)[44 45 54] and anorexia nervosa (n=2).[61 75] The anorexia case studies are two outliers in that the EoL condition was a direct result of the mental health issue for those individuals leading to chronic fractures and organ failure.[61 75] The vast majority of EoL conditions presented in the case study papers were cancer-related diagnoses, and organ failure (heart (n=3 (across four publications),[43 54 57 82] liver (n=1)[50] and renal (n=5)[58 66 73 77 81] made up the remainder.

### Critical appraisal scores
The quality of the studies varied overall (see online supplemental table 2). Twenty-five of the forty-two papers met seven or more criteria. In most cases, the single

criterion missing related to the description of diagnostic tests or assessment methods and their results. Eight of the forty-two papers met half or fewer of the eight quality criteria, but for the purposes of inclusivity all forty-two papers were included in the subsequent narrative synthesis.

### Thematic synthesis
These case studies report on key episodes, or critical junctures,[83] in the lives of people with SMI at the EoL. Five themes were identified centring on people with SMI presenting to health services and then disappearing again, or pivotal decisions with profound impact for patients and others. These themes are diagnostic delay and overshadowing; decisional capacity and dilemmas; medical futility; individuals and their networks; and care provision.

### Diagnostic delay and overshadowing
A consistent picture of delayed or late diagnosis and diagnostic overshadowing appears in case studies. Complex and enduring mental health problems may contribute to these issues. The problem of delay is signalled as arising from the mental illness of the individual who fails to recognise the seriousness of their plight, and presents to services only when their condition is advanced and treatment options limited.[52 59 60 64 71] For example, Ms A, an African American woman in her 30s with an unspecified psychosis presented with a 20 cm mass in her right breast adhering to the chest wall.[59] Investigations revealed a significant cancer with lymph node involvement with no metastases. Ms A was treated with chemotherapy and surgery, failed to attend for radiation treatment and considered new lesions as nothing more than 'haematoma'. She later presented in respiratory distress with large, presumably malignant effusions, asked not to be resuscitated and died soon after. The case study paper labels Ms A's difficulty with accepting her situation and similar behaviours as 'maladaptive denial'.

The tendency to impute the mental conditions of individuals as a source of late presentation, or indeed for deciding not to continue with treatment, is a recurring feature of case studies. Case studies indicate how diagnosis and treatment of serious life-threatening or life-limiting physical conditions are often complicated by untreated mental illness and the beliefs and behaviours arising from this.[59 60 64] Some case studies appear to merge conceptualisation of behavioural aspects of presentations with mental ill health and position these as problematic in the delivery of care and treatment with terms such as 'malingering', 'manipulative', 'dominance and aggression', 'demanding'[43 50 51 53] or problems for staff who 'struggled with how best to provide care'.[54]

Denial, or a lack of insight, is a possible complicating factor. It is not unknown for persons diagnosed with other serious conditions to deny the gravity of their situation, and for people with pre-existing mental health conditions it appears that this is no different.[59] Specific issues arise

in cases of PTSD, where 'the threat to life inherent in terminal illness may mimic the original trauma' leading to exacerbation of psychological symptoms associated with the condition including anxiety, anger, denial, avoidance and distrust of authority impeding medical adherence resulting in refusal of treatment.[54]

Individuals with long histories of pre-existing mental illness with regular access to healthcare professionals nevertheless experience delayed or late diagnosis of conditions that place them on the EoL trajectory.[20 47 55 65 78–80] For example, a man in his 60s with a longstanding diagnosis of schizophrenia, living in an adult foster home and under legal guardianship,[71] was seen fortnightly at a mental health clinic and attended a primary care provider to report a new onset cough. His examination was documented as benign, but 1 week later he reported hypotension and left-sided weakness. A detailed examination revealed advanced bladder cancer and multiple brain metastases not previously noted. The authors locate the problem in the mental condition, suggesting people with conditions such as schizophrenia do not willingly verbalise pain or related symptoms. It is, however, difficult to escape the conclusion that extended contact with healthcare services had failed to identify his condition.

### Decisional capacity and dilemmas

Decisional capacity of individuals to consent to treatment and/or to refuse treatments are reported and implicate professional dilemmas of determining the value of attempting curative treatments versus palliation.[46 48 50 55 59 60 73 74 81] It is argued[67] that clinicians have a particular duty to ensure that the interests of people with SMI are defended by offering medical treatment. While SMI can impair decisional capacity, this should not be assumed but instead thoroughly assessed.[67] In some examples, the conclusion is that the person retained capacity to determine their treatment choices, and these choices were then respected.[48]

Treatment refusal is highlighted numerous times.[48 52 55 56 60 63 66 67 70 72–74] Past refusals prompt treatment teams to impute future problems.[66] For example, a patient in their 80s with a 20-year history of a mastectomy for breast cancer and refusal of medical care, represented with a bleeding and ulcerating mass on the chest wall.[56] The case study reports that the patient was not a case for curative treatment despite no evidence of metastases.

Treatment refusal is reported as arising from psychiatric symptomatology such as fixed beliefs about damnation or that thoughts could be read by physicians, rather than due to capacity to understand and make decisions based on available information.[20]

Fluctuating mental capacity requiring multiple assessments[46 73] is reported. This can mean resort to the courts in treatment refusal[73] for declaration of lawfulness, being in the best interests of the patient, or of not imposing treatment notwithstanding the patient's inability to accept or refuse treatment. In one case of treatment refusal in a man with decisional incapacity, the medical ethics committee concluded that even with full decision-making capacity a person might reasonably refuse radical procedures due to risks involved and the deforming nature of surgery.[68]

The absence of previously declared wishes on life-saving treatments is a recurring issue and suggests one area for future intervention testing.[59 66 67 77] In these circumstances, teams seek agreement of a substitute decision-maker, such as a family member[66 67] sometimes with power of attorney. The patient may, however, indicate by their actions their refusal, for example, repeatedly removing life-saving treatments such as catheters in renal dialysis[77] or not agreeing to take medication.[59] Enforcing medical treatment when it is actively refused is not supported in this literature and may complicate future alliance building.[66]

### Medical futility

The concept of medical futility is invoked in case studies of people with anorexia nervosa, indicting how experience of this condition can exhaust the optimism of those doing the caring.[61] The language used, for example, 'refractory' and 'incurable' reflects this.[75] Ms A, a woman in her early 30s with a diagnosis of anorexia nervosa where there is use of palliation and referral to hospice care for the physical consequences of the psychiatric condition.[61] In this case, it is also reported that the option of an eating disorder treatment programme was eventually rejected on cost grounds, implying that perhaps the situation was less futile than indicated. Nevertheless, the treatment team and the ethics committee concluded, 'that her physical and psychiatric impairments were likely to lead to her death, despite any plausible attempts at aggressive intervention'.[61, p.373]

### Individuals and their networks

Support networks for people with SMI are crucial, and the absence of these is implicated in delayed treatment-seeking. Case studies refer to family involvement,[43 57 62 72 81 82] while others indicate the absence of such support.[50 52] The EoL condition in some circumstances appears to have led to the re-emergence of family support,[79] or that teams actively supported the person to reconnect with distant family.[48] For some, tensions in family involvement are reported[51 58] and in one example family dynamics are situated as the source of subsequent mental health-related troubles.[75] Case studies also identify family concerns for the individual with the EoL condition.[57 82] EoL care places additional demands on families, such as learning to manage symptoms,[44] and can lead to exhausted or burned-out family members.[61] One such demand arises from the absence of advance decisions, or where decisional capacity is in question. Families are then drawn into discussions on treatment and do not resuscitate decisions for which they are ill-prepared.[55 59 66 67 77] Families also express concern that the person's mental health problems are overlooked by the palliative care team.[72]

## Care provision

Case studies reveal issues in the provision of care for treatment teams, such as how to handle psychiatric presentations.[43 51 66 72] Case studies also report examples of what has worked in supporting people with mental health problems at the EoL, including: the building of rapport and trust especially in people with PTSD[44]; the use of music therapy at EoL[76]; having conversations about death[20 57 79 82]; initiating hospice at home[20 57 82]; and multi-disciplinary mental healthcare and palliative care being provided at home.[48]

Challenges reported include mental health staff being emotionally unprepared for caring for people who are terminally ill.[69] Where people with SMI receive EoL care in hospices palliative care staff experience strong emotions, such as anger and frustration.[44] Case study papers occasionally offer psychodynamic interpretations of staff/patient interactions and care.[43 58 69]

Case studies reporting the ongoing delivery of care to individuals with pre-existing SMI and subsequent EoL advocate the benefits of a transdisciplinary approach involving palliative care specialists, psychiatric specialists (preferably the team with a pre-existing relationship) and wider community members (eg, religious ministers).[9 20 47 48 61 67 75]

Liaison consultation and collaboration is reported as beneficial for hospice staff. Examples include psychiatric consultation,[53] psychologists advising on reducing environmental stress[44] and a hospice nurse being supported by a psychiatric nurse.[58] Mutual benefits of hospice and mental health nurses working together in EoL care, and the similarities in their work are reported. Hospice staff[72] and mental health staff[79] express interest in learning from each other when providing EoL care for people with mental health problems.

Where models of care are reported, these are small scale and do not indicate transferability or generalisability. Examples include treatment models, for example, DSA,[43] SPPC[44 45] and patient, provider, systems model.[20]

In almost every example, the place of care is positioned as one arising from the preferences of the individual. These include palliative care in mental health institutions[55 79 80]; hospice care for mental health patients[53 62 71 75]; acute hospital[9]; and home care.[48] Repeated transfer between settings related to psychiatric and/or physical symptomatology is also evident,[66] suggesting challenges to continuity of care.

Treatment challenges are reported in managing mental health-related medication alongside the provision of chemotherapy. Two case study papers report issues with the prescription of clozapine, as this is implicated in depressing white blood cell counts and could precipitate a life-threatening infection compounding the neutropenia associated with cytotoxic chemotherapy.[63 65] Treatment challenges also arise from people with SMI at the EoL absconding or being lost to follow-up.[64 74] A further challenge is in providing palliative care to patients who, despite careful explanation, continue to believe that they are not going to die.[61]

## DISCUSSION

This paper is the first to synthesise and present quality appraisal of published case studies of people with pre-existing SMI with EoL diagnoses. Case studies demonstrate the complexities and the ethical dilemmas associated with the provision of care to people with SMI at the EoL. They reveal stark challenges presented by delayed diagnosis, patients' fluctuating capacity to make decisions and a concern about the futility of treatment. Case studies also demonstrate the lack of preparedness of professionals to meet patients' multiple needs. For example, palliative care staff may lack mental health knowledge, and mental health staff often have little or no experience of palliative care. Despite this, professionals want to learn from each other across specialities to provide better care. Synthesised case study findings also show the dangers of ascribing delays in mobilising (or continuing with) palliative care services in response to patients' challenging presentations, or difficult behaviour.

Access to palliative care is variable, even for the general population in many countries with universal standards lacking, late or delayed referral and limited access to best care practices.[84] Recent reviews and synthesis of evidence describing clinical care in this field reveal similarly variable provision.[85–87] Research on EoL care for people with mental health problems is limited[35] and in the absence of high-quality intervention research, case studies provide an important source of knowledge. While there is much to learn from case studies, the reporting of clinical details is variable, thus limiting generalisation. A further challenge in synthesis arises in that almost all case studies come from high-income countries, limiting transferability to other settings.

Practitioners sometimes struggle with engaging individuals in difficult discussions about their care. However, people with SMI want these conversations.[88] Care provided across specialist boundaries and with the continuing support of already established therapeutic relationships is well supported. In providing care at the EoL for people with SMI practitioners must work to sustain these relationships, display compassion and sensitivity in difficult but necessary conversations, engage across disciplinary barriers to improve their understanding and establish preferences of the person in relation to EoL decisions. This finding has implications for the education and training of professionals across multiple disciplines, and must examine beliefs and attitudes towards mental ill health as well as assumptions about mental capacity and autonomy of people receiving services. In addition, legal contexts differ across and sometimes within countries, meaning the definition and understanding of issues such as mental capacity may vary.

The case studies synthesised here provide a range of clinical and non-clinical experiences of care for people

with EoL conditions in the context of SMI. These experiences point to a range of possible approaches for improving the prompt diagnosis and care of these individuals. In the absence of intervention studies, it remains unclear how robust these indications are for future clinical care.

Limitations of this review include our focus on papers in the English language, which may have led to the absence of important evidence published in other languages. The majority of case studies are from high-income countries and this limits generalisability to some extent, not least because there are likely to be contextual differences in health and social care systems across the globe. Nevertheless, EoL care for people with SMI is a global issue and further studies are required in different socioeconomic settings.

## CONCLUSION

This synthesis of case studies indicates that cross disciplinary care provided in the context of established therapeutic relationships and which values the preferences of individuals with SMI experiencing an EoL condition are supported in the absence of high-quality intervention research.

**Contributors** Conceived and designed the systematic review: MC, DE, SA, PG, MM, AM and BH. Conducted the literature search: MM and DE. Analysed the data: MC, DE, SA, PG, AM and BH. Wrote the paper: MC, DE, SA, PG, MM, AM and BH. Data interpretation and critical revision of manuscript: MC, DE, SA, PG, MM, AM and BH. All authors reviewed and approved the manuscript.

**Funding** This work was supported by the National Institute for Health Research (NIHR) Health Services and Delivery Research programme, grant number HS&DR 17/100/15 and will be published in full in Health Services and Delivery Research (https://www.journalslibrary.nihr.ac.uk/programmes/hsdr/1710015/#/). The views expressed are those of the authors and not necessarily those of the NIHR or the Department of Health and Social Care.

**Competing interests** None declared.

**Patient consent for publication** Not applicable.

**Ethics approval** This is a systematic review, no patient data were collected for this review and research ethics approval was not required. This study does not involve human participants.

**Provenance and peer review** Not commissioned; externally peer reviewed.

**Data availability statement** All data relevant to the study are included in the article or uploaded as supplementary information. All data of the current study are present in the main manuscript, figures, tables and online supplemental material.

**ORCID iDs**
Michael Coffey http://orcid.org/0000-0002-0380-4704
Deborah Edwards http://orcid.org/0000-0003-1885-9297
Sally Anstey http://orcid.org/0000-0003-2295-3761
Paul Gill http://orcid.org/0000-0003-4056-3230
Mala Mann http://orcid.org/0000-0002-2554-9265
Alan Meudell http://orcid.org/0000-0001-8138-4744
Ben Hannigan http://orcid.org/0000-0002-2512-6721

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
