## [Reviewer comments · BMJ Open]

ARTICLE DETAILS

TITLE (PROVISIONAL)	End of life care for people with severe mental illness: mixed methods systematic review and thematic synthesis of published case studies (the MENLOC study)
AUTHORS	Coffey, Michael; Edwards, Deborah; Anstey, Sally; Gill, Paul; Mann, Mala; Meudell, Alan; Hannigan, Ben

VERSION 1 – REVIEW

REVIEWER	White, Jacquie University of Hull, FHS
REVIEW RETURNED	23-Jul-2021

GENERAL COMMENTS	An important and well-presented paper that I was pleased to review as end-of-life care for people with SMI has received far too little attention to date in the published literature. There was one issue that concerned me that would strengthen the paper if it could be addressed before publication. I understand why a very broad definition of SMI was included in your published protocol and review to define your population and inclusion criteria, but I struggled to completely understand why the three papers reporting cases of people with a PTSD diagnosis, and the two reporting cases of people with an anorexia nervosa diagnosis had been included. The NIHR thematic review of support for people with SMI published in 2018 included two further considerations in their definition of SMI (beyond a diagnosis of schizophrenia, bi-polar disorder or other psychotic disorder); mental illness that results in significant disability in terms of day-to-day functioning and mental illness that has lasted for a significant duration, usually at least 2 years (p8). These latter two criteria could apply to the PTSD and AN case study papers you selected, with the possible exception of the case where acute PTSD symptoms were described as (re-)emerging due to the receipt of the terminal diagnosis and treatment. I think it is important to be explicit about why these five cases were included in your review and any limitations this then placed on your results. PTSD and AN are not usually included in definitions of SMI and could be considered completely different populations. For example mortality in adults with a diagnosis of AN has been described as the highest of any psychiatric disorder. You did describe the AN cases as outliers (Attributes of service users – p11) but there was no further discussion of the implications of this difference. There was no discussion of the implications (or limitations) of the inclusion of the PTSD cases. People deserve to receive good quality end of life care whatever their diagnosis and different definitions of SMI have been used in
---

	research and policy and have changed over the years (e.g. to recognise the needs of people with major depressive disorder and those with personality and behavioural disorders) but I think it is important to be more transparent about the inclusion of the PTSD and AN papers in your study because readers may want to duplicate your review, or use it to inform a protocol for future intervention studies.
--	---

REVIEWER	Wilson, Rebecca King's College London, Cicely Saunders Institute, Division of Palliative Care, Policy & Rehabilitation
REVIEW RETURNED	26-Jul-2021

GENERAL COMMENTS	This is a well conducted and well written systematic review. I have some suggestions that I hope you will find useful. Introduction  - There have been several recent reviews in this area, how was yours different and what was the rationale for conducting it? How did you anticipate it would add to the literature? Methods  - Line 45 – PRISMA is a reporting framework, not guidelines for conducting a review, suggest rewording - Re: inclusion criteria - including only cancer and organ failure related palliative cases excludes some significant patient groups eligible for palliative care, eg, frailty, degenerative neurological conditions. If you chose to exclude all other patients groups, I think you need to justify why under the exclusion criteria - How are you defining case studies? This needs expanding so it is clear what studies will be included in/excluded from your review, are they exclusively n=1 studies? - Where there any language restrictions? If no exclusion criteria were applied for language, were any excluded based on language? (I couldn't find Appendix 14 with reasons for exclusion) - There's no real information on the tool you used for critical appraisal, why was this tool chosen and what elements of the studies did it consider? Results  - Line 3-5 Page 9 – This is quite confusing as it reports studies that are not included, as they are not case studies? Why are they here? - Line 40 Page 11 – I'm not sure 'elide' is what you mean, highlight? - Line 25 Page 13 – says 'recurring issue' but there is only one reference? Either explain why it is recurring, and add more references, or reword Discussion  - The discussion is a bit scant, I was left thinking what the contribution made to the literature is by this manuscript. I think it needs an 'implications' section and 'strengths and limitations'. And also discuss how this study fits within the wider evidence base. Yes the evidence is limited but it is growing. There are several recent reviews that I think you could discuss your review in relation to (see reference list below). - Line 7-9 Page 16 – suggest rewording to "palliative care staff may" or "often lack knowledge"
--

	- Line 23-25 Page 16 – in a recent review using more quantitative synthesis of the evidence (Wilson et al 2020), describing/summarising clinical care was similarly variable/disparate, I suggest drawing parallels to strengthen your results - Line 34-36 Page 16 – supported by Elie et al (2018), suggest citing References Donald, E. E., & Stajduhar, K. I. (2019). A scoping review of palliative care for persons with severe persistent mental illness. Palliative & supportive care, 17(4), 479-487. French, M., Keegan, T., Anestis, E., & Preston, N. (2021). Exploring Socioeconomic Inequities in Access to Palliative and End-of-life Care in the Uk: A Mixed Methods Narrative Synthesis. Wilson, R., Heggul, N., Higginson, I. J., & Gao, W. (2020). End-of-life care and place of death in adults with serious mental illness: A systematic review and narrative synthesis. Palliative Medicine, 34(1), 49–68. Elie, D., Marino, A., Torres-Platas, S. G., Noohi, S., Semeniuk, T., Segal, M., ... & Rej, S. (2018). End-Of-Life care preferences in patients with severe and persistent mental illness and chronic medical conditions: a comparative cross-sectional study. The American Journal of Geriatric Psychiatry, 26(1), 89-97.
--	--

REVIEWER	Brennan, Gearoid University of Stirling, Faculty of Health Sciences & Sport
REVIEW RETURNED	03-Aug-2021

GENERAL COMMENTS	Thank you for submitting a most interesting systematic review and thematic synthesis. The topic is so pertinent. A real strength of this review is that it very nicely paints the landscape so that indeed intervention studies can be developed on the back of a thorough and sound review of the existing literature.  1. This review had a clear focus, specifically the aims around organisation, provision and receipt of End-of-life care. 2. The search and synthesis have evidently been conducted in a robust and transparent manner. There was a broad and varied range of databases accessed. 3. There was a clear focus on accessing case studies. There is a clear justification for this, and it makes for good clinical utility. 4. I liked that the authors have clearly acknowledged the geographical limitations of studies coming from high-income regions. This is important, as the reduced life expectancy and high levels of co-morbidities are a global issue. However, the causes are somewhat different depending on the region. The authors could strengthen their review by adding a line making this point expressly clear. It is a global issue, and it is likely end-of-life care for people with SMI is impacted on this global scale. I would suggest that accessing the Lancet Psychiatry's Blueprint for physical health (Firth et al 2019) may help position the review in this context. Though it is worth noting that the blueprint does not mention EoL or palliative care. https://www.thelancet.com/journals/lanpsy/article/PIIS2215-0366(19)30132-4/fulltext
--

	5. There is a clear definition given by what the authors mean by end-of-life care. 6. It is a well written and well referenced manuscript 7. Please state within the main article that you used JBI tool for critical appraisal. I know it is included in the supplementary material but it is not clearly stated within the main body that this was used or the rationale for this tool over another one. 8. The findings were very interesting and probably not very surprising. Re: Decisional capacity. Again, as studies were drawn from an international base, the legal framework in this area will differ greatly. For example, even within the UK the legislation around capacity differs between Scotland and England. This is to say nothing of the Mental Health legislation and Scotland's 'Significantly Impaired Decision-making ability (SIDMA) which is different to capacity. I think the authors need to make this point more explicit- particularly if it will inform intervention work as the contexts are so diverse and varying internationally. 9. It is clear how the authors arrived at suggestions for future work- in particular, future decision making/living will is a much needed intervention. 10. The lack of mobilising Palliative care is not unique to those with SMI but also recognised as an issue in the rest of the population. This should be acknowledged within the manuscript. It sparks joy for me when I get to review manuscripts such as this. Such topics are so under researched, so it is fantastic to see this work going on, so that the inequalities that people with SMI are being addressed and published in journals such as BMJ Open. Thank you!
--	--

VERSION 1 – AUTHOR RESPONSE

Reviewer: 1 Dr. Jacquie White, University of Hull Comments to the Author: An important and well-presented paper that I was pleased to review as end-of-life care for people with SMI has received far too little attention to date in the published literature. There was one issue that concerned me that would strengthen the paper if it could be addressed before publication.	Thank you
I understand why a very broad definition of SMI was included in your published protocol and review to define your population and inclusion criteria, but I struggled to completely understand why the three papers reporting cases of people with a PTSD diagnosis, and the two reporting cases of people with an anorexia nervosa diagnosis had been included. The NIHR thematic review	As you note that NIHR review allow for duration and disability as additional criteria and this follows from the definition applied in the Building Bridges report. The decision on inclusion was made with our steering group and the

of support for people with SMI published in 2018 included two further considerations in their definition of SMI (beyond a diagnosis of schizophrenia, bi-polar disorder or other psychotic disorder); mental illness that results in significant disability in terms of day-to-day functioning and mental illness that has lasted for a significant duration, usually at least 2 years (p8). These latter two criteria could apply to the PTSD and AN case study papers you selected, with the possible exception of the case where acute PTSD symptoms were described as (re-)emerging due to the receipt of the terminal diagnosis and treatment. I think it is important to be explicit about why these five cases were included in your review and any limitations this then placed on your results. PTSD and AN are not usually included in definitions of SMI and could be considered completely different populations . For example mortality in adults with a diagnosis of AN has been described as the highest of any psychiatric disorder. You did describe the AN cases as outliers (Attributes of service users – p11) but there was no further discussion of the implications of this difference. There was no discussion of the implications (or limitations) of the inclusion of the PTSD cases.	intention was to not expressly focus on diagnosis but to examine papers for evidence of disability and duration of mental ill health. PTSD is an enduring condition and recurring in the sense that it can appear to resolve but re-emerge in traumatic times as we see in these case studies. All included case studies also met our requirement for these conditions to be pre-existing and not arising solely as a result of the end-of-life diagnosis. We don't believe there are any limitations arising from this on our results and we have attempted to be clear on this when reporting. To direct readers to the rationale for our inclusion decisions and the related discussions we have inserted a line citing our main NIHR report which will appear in due course and acknowledges the challenges of SMI definitions and the imprecise nature and broadness of these.
People deserve to receive good quality end of life care whatever their diagnosis and different definitions of SMI have been used in research and policy and have changed over the years (e.g. to recognise the needs of people with major depressive disorder and those with personality and behavioural disorders) but I think it is important to be more transparent about the inclusion of the PTSD and AN papers in your study because readers may want to duplicate your review, or use it to inform a protocol for future intervention studies.	We entirely agree with your point and were motivated to design this study based on conversations with service users with relevant experiences in this area. We have added a line to the inclusion criteria making clear that these conditions were included and cite our main report for the full detail on this decision.
Reviewer: 2 Dr. Rebecca Wilson, King's College London Comments to the Author: This is a well conducted and well written systematic review. I have some suggestions that I hope you will find useful.	Thank you
Introduction - There have been several recent reviews in this area, how was yours different and what was the rationale for conducting it? How did you anticipate it would add to the literature?	We have added a clarifying sentence to the introduction to address this point. We believe our review is distinguished from previous work as we included case studies and a review of policy papers (in the main report).

Methods - Line 45 – PRISMA is a reporting framework, not guidelines for conducting a review, suggest rewording	Thank you, yes, sentence has been reworded although the original wording is that suggested in authors guidelines for this journal.
- Re: inclusion criteria - including only cancer and organ failure related palliative cases excludes some significant patient groups eligible for palliative care, eg, frailty, degenerative neurological conditions. If you chose to exclude all other patients groups, I think you need to justify why under the exclusion criteria	Thank you for this suggestion. We have added an explanation to this section. Our interest was to achieve a focused review. Our discussions with our steering group indicated that adding in these other groups would reduce the specificity of our work. Instead we plan separate work related to the groups excluded in this review, should we successfully secure funding.
- How are you defining case studies? This needs expanding so it is clear what studies will be included in/excluded from your review, are they exclusively n=1 studies?	For the purposes of this review case studies were all descriptions of the care and experiences of individuals with pre-existing SMI and EoL diagnosis identified via our search strategy and were included. Case studies were excluded if no SMI condition was identified, if no EoL condition was identified or if the mental illness was not pre-existing.
- Where there any language restrictions? If no exclusion criteria were applied for language, were any excluded based on language? (I couldn't find Appendix 14 with reasons for exclusion)	English language only papers were searched and this is indicated in the original manuscript on page 8 line 7.
- There's no real information on the tool you used for critical appraisal, why was this tool chosen and what elements of the studies did it consider?	Thank you, we used the JBI case report checklist. This is noted in the original manuscript on page 9 lines 36-42. The scoring of studies is in supplementary file 5. We have added an additional sentence to indicate what the items in the scale seek to appraise.
Results - Line 3-5 Page 9 – This is quite confusing as it reports studies that are not included, as they are not case studies? Why are they here?	We included these for completeness but recognise they might add to confusion and have now removed this from here and from the abstract.

- Line 40 Page 11 – I'm not sure 'elide' is what you mean, highlight?	Not highlight, some case studies appeared to make assumptions about behavioural presentations as if they might be considered synonymous with mental ill health. We have revised 'elide' for clarity to 'merge'
- Line 25 Page 13 – says 'recurring issue' but there is only one reference? Either explain why it is recurring, and add more references, or reword	We suspect it is recurring because just like the general public, not everyone declares their wishes with regard to life-saving treatments. In the case of mental illness however practitioners show concern that they might cause undue upset but also appear to readily assume lack of capacity. Citations have been added to this sentence for studies where this issue was raised.
Discussion - The discussion is a bit scant, I was left thinking what the contribution made to the literature is by this manuscript. I think it needs an 'implications' section and 'strengths and limitations'. And also discuss how this study fits within the wider evidence base. Yes the evidence is limited but it is growing. There are several recent reviews that I think you could discuss your review in relation to (see reference list below). - Line 34-36 Page 16 – supported by Elie et al (2018), suggest citing	We're slightly worried about the word count if we add in too much extra content such as suggested here. We value the suggestion however and have added some sentences in the discussion to pick up implications and limitations. We have taken your suggestions and added the references supplied, many of which are included in our main report. Thank you for the reminder on this paper which we have now included to support the point.
- Line 23-25 Page 16 – in a recent review using more quantitative synthesis of the evidence (Wilson et al 2020), describing/summarising clinical care was similarly variable/disperate, I suggest drawing parallels to strengthen your	Thank you for this useful point, we have added it to the discussion and included our own recently published review in addition.

results	
- Line 7-9 Page 16 – suggest rewording to “palliative care staff may” or “often lack knowledge”	Thank you, yes, your wording is more generous, and we have made this revision.
References Donald, E. E., & Stajduhar, K. I. (2019). A scoping review of palliative care for persons with severe persistent mental illness. Palliative & supportive care, 17(4), 479-487. French, M., Keegan, T., Anestis, E., & Preston, N. (2021). Exploring Socioeconomic Inequities in Access to Palliative and End-of-life Care in the Uk: A Mixed Methods Narrative Synthesis. Wilson, R., Heggul, N., Higginson, I. J., & Gao, W. (2020). End-of-life care and place of death in adults with serious mental illness: A systematic review and narrative synthesis. Palliative Medicine, 34(1), 49–68. Elie, D., Marino, A., Torres-Platas, S. G., Noohi, S., Semeniuk, T., Segal, M., ... & Rej, S. (2018). End-Of-Life care preferences in patients with severe and persistent mental illness and chronic medical conditions: a comparative cross-sectional study. The American Journal of Geriatric Psychiatry, 26(1), 89-97.	All of these citations are now included in the paper with the exception of French et al as it is not yet an accepted paper.
Reviewer: 3 Dr. Gearoid Brennan, University of Stirling Comments to the Author: Thank you for submitting a most interesting systematic review and thematic synthesis. The topic is so pertinent. A real strength of this review is that it very nicely paints the landscape so that indeed intervention studies can be developed on the back of a through and sound review of the existing literature.	Thank you
1. This review had a clear focus, specifically the aims around organisation, provision and receipt of End-of-life care.	Again, thanks for this comment.
2. The search and synthesis have evidently been conducted in a robust and transparent manner. There was a broad and varied range of databases accessed.	Thank you
3. There was a clear focus on accessing case studies. There is a clear justification for this, and it makes for good clinical utility.	With limited evidence they do give us very usual insights and its reassuring that you have picked up on this.

4. I liked that the authors have clearly acknowledged the geographical limitations of studies coming from high-income regions. This is important, as the reduced life expectancy and high levels of co-morbidities are a global issue. However, the causes are somewhat different depending on the region. The authors could strengthen their review by adding a line making this point expressly clear. It is a global issue, and it is likely end-of-life care for people with SMI is impacted on this global scale. I would suggest that accessing the Lancet Psychiatry's Blueprint for physical health (Firth et al 2019) may help position the review in this context. Though it is worth noting that the blueprint does not mention EoL or palliative care. https://www.thelancet.com/journals/lanpsy/article/PIIS2215-0366(19)30132-4/fulltext	Thanks for this point, we have picked this up with a line in the discussion following on from acknowledging limitations. We have not added the Lancet citation for the reason you note.
5. There is a clear definition given by what the authors mean by end-of-life care.	Thank you
6. It is a well written and well referenced manuscript	Thank you
7. Please state within the main article that you used JBI tool for critical appraisal. I know it is included in the supplementary material but it is not clearly stated within the main body that this was used or the rationale for this tool over another one.	You may have missed this in the original manuscript, it is mentioned under the heading 'Assessment of methodological quality'
8. The findings were very interesting and probably not very surprising. Re: Decisional capacity. Again, as studies were drawn from an international base, the legal framework in this area will differ greatly. For example, even within the UK the legislation around capacity differs between Scotland and England. This is to say nothing of the Mental Health legislation and Scotland's 'Significantly Impaired Decision-making ability (SIDMA) which is different to capacity. I think the authors need to make this point more explicit- particularly if it will inform intervention work as the contexts are so diverse and varying internationally.	Thank you for this suggestion, we have added a line to make this point as a follow-on from likely implications in the discussion section.
9. It is clear how the authors arrived at suggestions for future work- in particular, future decision making/living will is a much needed intervention.	Thank you
10. The lack of mobilising Palliative care is not unique to those with SMI but also recognised as an issue in the rest of the population. This should be acknowledged within the manuscript. It sparks joy for me when I get to review manuscripts such as this. Such topics are so under researched, so it is fantastic to see this work going on, so that the inequalities that people with SMI are	Thanks for the suggestion for this important point. We have added a sentence to the discussion with a relevant supporting citation.

being addressed and published in journals such as BMJ Open. Thank you!	
---	--

VERSION 2 – REVIEW

REVIEWER	Wilson, Rebecca King's College London, Cicely Saunders Institute, Division of Palliative Care, Policy & Rehabilitation
REVIEW RETURNED	11-Oct-2021
GENERAL COMMENTS	The authors have sufficiently addressed all comments and I would recommend accepting this manuscript.